# Rapid Identification of Rainbow Trout Adulteration in Atlantic Salmon by Raman Spectroscopy Combined with Machine Learning

**DOI:** 10.3390/molecules24152851

**Published:** 2019-08-06

**Authors:** Zeling Chen, Ting Wu, Cheng Xiang, Xiaoyan Xu, Xingguo Tian

**Affiliations:** 1College of Food, South China Agricultural University, Guangzhou 510642, China; 2School of Information Science and Technology, Zhongkai University of Agriculture and Engineering, Guangzhou 510225, China; 3New Rural Development Research Institute, South China Agricultural University, Guangzhou 510225, China

**Keywords:** Atlantic salmon, adulteration, Raman spectroscopy, machine learning

## Abstract

This study intends to evaluate the utilization potential of the combined Raman spectroscopy and machine learning approach to quickly identify the rainbow trout adulteration in Atlantic salmon. The adulterated samples contained various concentrations (0–100% *w*/*w* at 10% intervals) of rainbow trout mixed into Atlantic salmon. Spectral preprocessing methods, such as first derivative, second derivative, multiple scattering correction (MSC), and standard normal variate, were employed. Unsupervised algorithms, such as recursive feature elimination, genetic algorithm (GA), and simulated annealing, and supervised K-means clustering (KM) algorithm were used for selecting important spectral bands to reduce the spectral complexity and improve the model stability. Finally, the performances of various machine learning models, including linear regression, nonlinear regression, regression tree, and rule-based models, were verified and compared. The results denoted that the developed GA–KM–Cubist machine learning model achieved satisfactory results based on MSC preprocessing. The determination coefficient (R^2^) and root mean square error of prediction sets (RMSEP) in the test sets were 0.87 and 10.93, respectively. These results indicate that Raman spectroscopy can be used as an effective Atlantic salmon adulteration identification method; further, the developed model can be used for quantitatively analyzing the rainbow trout adulteration in Atlantic salmon.

## 1. Introduction

Atlantic salmon (Salmo salar) has attracted consumer interest because of its unique taste and rich nutritional value. Even though there is a huge demand for Atlantic salmon in the Chinese market, imported Atlantic salmon is often in short supply. Under these circumstances, rainbow trout (Oncorhynchus mykiss) is often used to imitate or adulterate Atlantic salmon meat or products. Rainbow trout is considerably less expensive than Atlantic salmon but looks similar, making it difficult for consumers to distinguish between the two. Adulterated Atlantic salmon meat not only infringes the legitimate rights and interests of consumers but also causes serious food safety problems, leading to widespread concern among consumers, producers, retailers, and food regulatory agencies.

Therefore, an accurate and expedient method is required for identifying Atlantic salmon adulteration. Traditional meat identification methods have mainly used enzyme-linked immunosorbent assays, deoxyribonucleic acid (DNA) [1,2,3,4,5,6], proteome [7,8,9], and triacylglycerol-based analytical techniques [10]. Although these methods have proved to be accurate, they exhibit long processing times and complicated technologies, making them unsuitable for rapid on-site detection under market supervision environments.

Among the emerging detection technologies, infrared spectroscopy, hyperspectral imaging, and Raman spectroscopy have been proved to be valuable for characterizing the chemical structure of the meat molecules [11,12]. Further, these techniques have been regularly employed to assess the quality [13,14,15], safety [16,17,18], and classification [19,20] of the meat products [21,22,23]. Among them, Raman spectroscopy has been considered to be a promising meat detection tool because it is a non-invasive, fast, and convenient technique that requires little to no sample pretreatment [20,24,25,26,27,28]. Recently, especially after the “horse meat storm” incident [29], Raman spectroscopy has gradually been used for identifying meat adulteration [29,30,31,32,33,34,35,36,37,38,39]. For example, Boyaci et al., used Raman spectroscopy to scan beef and horse fat samples and used the principle component analysis method for performing data processing and modeling [29]. The results denoted that this model could distinguish among different horse meat contents (25%, 50%, and 75%) in beef samples. Zhou Yaling et al. [32] used Raman spectroscopy combined with chemometrics to establish a discriminant model based on support vector regression. The results showed that the model was able to accurately identify the adulterated beef stuffing with different proportions of chicken meat.

In previous studies, Raman spectroscopy has often been used to identify sausages [38], fish [23], beef [24], and other common meat. In these examples, the full potential of Raman spectroscopy could not be evaluated because the minimum detection limit of the employed model was often large. In addition, Raman spectroscopy has mostly been used for qualitative discrimination, which does not lend itself to predicting and quantifing the adulteration level in meat. Furthermore, to the best of our knowledge, Raman spectroscopy has not been used to identify Atlantic salmon adulteration. Therefore, we were interested in evaluating the ability of Raman spectroscopy to detect low adulteration levels in Atlantic salmon meat. This study discloses our efforts in developing a convenient, high-sensitivity, low-cost Atlantic salmon meat adulteration detection technology based on Raman spectroscopy combined with a stable, fast, and accurate machine learning model.

The objectives of this study are as follows: (1) to evaluate the effectiveness of Raman spectroscopy in identifying the adulteration of Atlantic salmon meat and to develop a portable Raman spectroscopy and machine learning method to quickly identify rainbow trout adulteration in Atlantic salmon; (2) to compare the four pretreatment methods [first derivation (FD), second derivation (SD), multiple scattering correction (MSC), and standard normal variate (SNV)] for obtaining the best pretreatment method; (3) to combine the supervised (RFE, GA, and SA) and unsupervised (KM) dimensionality reduction and variable selection methods to find the best characteristic wavelengths; and (4) to test the performance of mainstream machine learning models, such as linear regression, nonlinear regression, regression trees, and rule-based methods, in case of Atlantic salmon adulteration.

## 2. Results

### 2.1. Raman Spectral Analyses

As the Raman spectra of lower than 500 cm^−1^ and higher than 2000 cm^−1^ obtained in various regions exhibited a large amount of spectral noise and no obvious absorption peaks, bands with rich spectral information (500–2000 cm^−1^) were selected for conducting spectral analysis. The Raman spectra of Atlantic salmon and rainbow trout fat within these regions are depicted in Figure 1.

To eliminate the effects of baseline drift and scattering distortion on the spectra and to compare the spectral differences among the two fish, preprocessing methods, including baseline correction and MSC, were conducted, as depicted in Figure 1a. Some significant differences were observed between the intensities of the spectral absorption peaks of the two fish. The mean and standard deviation spectra of salmon and rainbow trout have been depicted in Figure 1b. Eight overlapping peaks with different intensities (with the exception of 1748 cm^−1^) were identified in two spectra; the peaks associated with the rainbow trout were characterized as the stronger of the two. These peak intensity differences formed the basis for distinguishing between rainbow trout and Atlantic salmon. The component functional groups corresponding to the Raman peaks of the two fish fats were analyzed and have been provided in Table 1.

As presented in Table 1, the peak at 1748 cm^−1^ is weak in strength and is attributed to the C=O ester stretching mode (C=O). The peak at 1659 cm^−1^ corresponds to a Z-alkene, ν(C=C), in the fatty acid chain, whereas the strong peak at 1441 cm^−1^ corresponds to the C–H bending stretching modes. The peak at 1303 cm^−1^ is attributed to the CH_2_ twisting modes (C–H), the peak at 1268 cm^−1^ is due to the Z conformation stretching modes (=CH) from the unsaturated fatty acids, the peaks at 1079 cm^−1^ and 872 cm^−1^ are due to gauche C–C stretching vibrations (C–C), and the peak at 974 cm^−1^ is caused by the bending vibration of trans (=CH). These peaks exhibited medium strength. The eight characteristic peaks have also been reported as common features in the Raman spectra of edible oils below 2000 cm^−1^ [40,41,42], and the peak positions and intensities of different fatty acids have been observed to be slightly different [43].

The Raman spectra in case of Atlantic salmon with different proportions of rainbow trout adulteration are depicted in Figure 2.

The Raman spectra of the adulterated Atlantic salmon fat were very similar, with characteristic absorption peaks being observed at 1748, 1659, 1441, 1303, 1268, 1079, 974, and 872 cm^−1^. The Raman peak intensity increased with increasing amounts of rainbow trout meat, enabling the Raman spectra of the different Atlantic salmon meat samples to be distinguished. These differences in absorption peaks provided the basis for further model development.

### 2.2. Preprocessing Analysis

The spectra of the Atlantic salmon samples obtained using different pretreatment methods, such as baseline correction, MSC, SNV, FD, and SD, are depicted in Figure 3a–f.

To compare the effects of different pretreatment methods, a partial least squares regression (PLSR) model was used to evaluate the results of the pretreatment methods. The experiment conducted without a pretreatment step was used as a reference. The results are presented in Table 2.

The model performance was the highest when the principal component number of PLSR modeling was 10. While observing the test sets, RMSEP and R^2^ could reach values of 17.27 and 0.70 in case of the usage of raw spectral modeling, respectively. In contrast, the RMSEP values obtained using FD and SD were 23.10 and 30.15 and R^2^ values were 0.48 and 0.12, respectively. These results demonstrated that FD and SD modeling were less effective than raw spectral modeling. The FD and SD methods amplified the noise in the spectra, which can explain the poor performance of these models. Further studies using MSC and SNV revealed that both the methods could achieve better results when compared with the original spectra, i.e., the resulting RMSEP was smaller and R^2^ was larger. The two methods also eliminated the scattering effect that negatively influenced the spectral data.

By comparing the two methods in cases in which they performed similarly, the number of principal components required for SNV modeling was observed to be larger than that required for MSC. Therefore, the MSC modeling performance could be considered to be slightly better than SNV. Further, while comparing different machine learning modeling methods in this study, MSC was the only method employed for spectral preprocessing.

### 2.3. Important Spectral Band Selection

In this study, different supervised methods (RFE, GA, and SA) and unsupervised methods (KM) were combined to reduce the dimension of spectral wavelengths, and optimal bands were selected (Table 3).

The results presented in Table 3 denote that the performances of the three methods were relatively similar and not distinct from those of the full-spectra model. However, the number of required spectral bands was considerably reduced in comparison with that in the full-spectra method, improving both the efficiency and stability of the model. Among them, GA–KM was considered to be the best method; the required wavelengths of the model were considerably reduced from 882 to 431, and this method exhibited improved prediction performance (R^2^ = 0.81, RMSEP = 13.34%).

### 2.4. Results of the Cubist Model

After applying the MSC pretreatment method and selecting the optimal feature bands by GA–KM, the Cubist model was established to identify adulterated Atlantic salmon samples containing different proportions of rainbow trout. To optimize the Cubist model, different sizes and numbers of committees and instances in the model were examined. The cross-validation curve of the Cubist model is presented in Figure 4.

Regardless of the number of instances, the error significantly decreased as the commits gradually increased to 10. While increasing the number of commits from 10 to 20, the error only slightly decreased. When the Cubist model used 20 commits and 5 instances, the error was the smallest and the modeling effect was the largest. Furthermore, when the number of instances was too low or too high, the performance of the model would decrease. The adulteration ratio of Atlantic salmon was predicted based on the aforementioned parameters, and the results are presented in Figure 5.

The RMSE in the calibration sets was 12.67, and R^2^ was 0.84; these values were 10.93 and 0.87, respectively, in the test sets. The experimental data exhibited a high degree of agreement with the predicted data, and the modeling performance was good, suggesting that this technique could be used to quickly identify Atlantic salmon adulteration.

## 3. Discussion

To denote the advantages of the Cubist algorithm with respect to model prediction, 13 types of machine learning methods and the PLSR method were used to model the selected spectral bands. The results are summarized in Table 4. For the test sets, the Cubist model was observed to have the smallest RMSEP (10.98) followed by PLSR (13.34). This result indicated that the modeling performance of the linear regression model (Cubist and PLS) may be better than those of other models. One explanation for this result is that adulteration using the rainbow trout followed a linear relation in the Raman spectra; as the adulteration ratio increased, the peak strength of the Raman spectra increased. The RMSEP of the Cubist method was much smaller than that of the remaining models, which could have been due to the Cubist method being a rule-based model. The model tree leaf node was a linear regression model, and the regression equation modeling on the node was more flexible and more accurate when compared with the other regression models [44]. Furthermore, complex linear regression models did not yield better performances. The RMSEP of Glmboost, Enet, Ridge, and Rqlasso were not as suitable as that of the PLS model. The modeling performances obtained using nonlinear models, such as random forests and neural networks, were even worse than those of the complex linear models. This indicated that the more complex linear or nonlinear models were not globally optimal when the number of samples was not sufficiently large and that they were prone to over-fitting, leading to a decrease in the accuracy of the model. In summary, the modeling performance of the linear model was generally better than those of the other models, and there was a linear relation between the rainbow trout adulteration and peak intensity of the Raman spectra. The Cubist model exhibited the best modeling performance and was combined with Raman spectroscopy to develop a new technique for identifying Atlantic salmon adulteration.

## 4. Materials and Methods

### 4.1. Sample Preparation

Different amounts of rainbow trout were added to Atlantic salmon to create adulterated samples. To expand the sample diversity and improve the credibility of the experimental results, Atlantic salmon was obtained from different batches, at different times, and from different regions (Denmark, Scotland, Chile, and Norway). To ensure the authenticity of the samples, import-certified Atlantic salmon stores were selected. Danish Atlantic salmon meat was purchased from Hippo Fresh Food in Guangzhou, China; Chilean Atlantic salmon meat was purchased from Jingdong Supermarket in Guangzhou, China; Scottish Atlantic salmon meat was purchased from Haidi Wang Fresh Seafood in Shanghai, China; and Norwegian Atlantic salmon meat was purchased from the Yuesheng official store in Shenzhen, China. Rainbow trout was also purchased from different regions and stores. Qinghai rainbow trout was purchased from the Tmall Longyangxia store in Gonghe, China; Gansu rainbow trout was purchased from the Tmall Shangzhi store in Lanzhou, China; Shandong rainbow trout was purchased from the Laoshan ecological farm in the Taobao store in Qingdao, China; and Liaoning rainbow trout was purchased from the Tmall supermarket in Benxi, China.

The Atlantic salmon meat obtained from four regions (Denmark, Chile, Scotland, and Norway) and the rainbow trout obtained from Qinghai, Gansu, Shandong, and Liaoning were all crushed using a grinder in KRUPS, Germany. Further, the Atlantic salmon and rainbow trout were mixed according to the following weight percentages: 0%, 10%, 20%, 30%, 40%, 50%, 60%, 70%, 80%, 90%, and 100% (*w*/*w*). Each of the 176 mixed samples weighed 50 g, and 2–3 parallel samples were prepared for each gradient, affording 516 prepared samples. Each sample was subsequently homogenized using a meat grinder and centrifuged at a speed of 10,000 rpm and a centrifugation time of 5 min. After centrifugation, the upper layer of an oily material was pipetted into a chemical reaction plate for conducting the Raman spectroscopy measurements.

### 4.2. Raman Spectral Data Measurements

A portable Raman spectrometer (FoodDefend RM, Thermo Fisher, Waltham, MA, USA) was used to collect the Raman spectra. The Raman system was equipped with a laser excited at 785 nm. When scanning, the laser power source was set to 250 mW, and the spectral range was 250–2500 cm^−1^. The spectral resolution was 7 cm^−1^, the exposure time was 5 ms, the scanning delay was 60 s, and the operating temperature was set to 30 °C to achieve the optimal Raman peaks. To eliminate noise and ensure data repeatability, each sample was scanned thrice; the average values were used as the sample spectra.

### 4.3. Spectral Pretreatment

Before modeling, spectral preprocessing was required to reduce noise or to eliminate random and systematic changes in the data [26]. The four preprocessing methods (FD, SD, MSC, and SNV) had different effects on the spectral data. For example, FD was used to remove the baselines, SD was used to remove the baselines and linear trends [45], and MSC and SNV were typically used to eliminate the unwanted scattering effects [46]. In this study, PLSR was used to model the same spectral data, and the effects of four pretreatment methods were evaluated for conducting the Atlantic salmon contamination analyses.

### 4.4. Analytical Methods

#### 4.4.1. Spectral Band Selection Methods

Selection of important spectral bands was critical to reducing the high dimensionality of the spectral data and increasing the processing speed [16]. Some researchers used unsupervised methods, such as clustering algorithms, for conducting feature selection [47,48]. As the clustering method has been shown to result in the selection of irrelevant features [47,48,49], irrelevant features were deleted before feature clustering. Some supervisory feature selection methods, such as RFE [50], GA [51], and SA [52], have been shown to be effective approaches for removing unrelated features. Furthermore, we attempted to combine the supervised and unsupervised methods for performing dimensionality reduction and variable selection. Firstly, the RFE, GA, and SA algorithms were used to remove the uncorrelated wavelengths and select the relevant characteristic bands. Then, KM [53,54,55] was used to optimize the feature wavelengths, and PLSR was used to monitor the modeling error of these selections. The optimal feature wavelengths could be obtained based on these data.

#### 4.4.2. Modeling Methods

Certain machine learning models have proven to be effective for identifying food adulteration [56,57,58]. In this study, the applicability of several machine learning models for predicting Atlantic salmon adulteration were evaluated. The R language was used for modeling, and a total of 14 mainstream machine learning algorithms, including linear regression models, nonlinear regression models, tree-based, and rule-based models, were used for training and testing. Linear regression models included PLSR, the boosted generalized linear model (Glmboost) [59], Elasticnet regression (Enet) [60], ridge regression (Ridge) [61], quantile regression with LASSO penalty (Rqlasso) [62], multi-step adaptive MCP-net (Msaene) [63], quantile random forest (Qrf) [64], parallel random forest (parRF) [65], random forest (Rf) [66], k-nearest neighbors (Kknn) [67], and multivariate adaptive regression spline (Earth) [68]. The tree-based models include conditional inference tree (ctree) [69] and extreme gradient boosting (xgbTree) [70]. The Cubist model (Cubist) [71] was the only rule-based model considered in this study. The selected wavelength sets and adulteration levels were used as the input and output variables, respectively, for the model, and the input and output data and other conditions were observed to be consistent while evaluating and comparing the model performance. A random sampling method was selected to divide the data sets into two subsets: training data (75%) and test data (25%). When modeling, 10-fold cross-validation was used, and five training times were repeated and averaged for the final results. The aforementioned process was implemented using the R language Caret package. 

### 4.5. Model Evaluation

The determination coefficient (R^2^), root mean square error of calibration sets (RMSEC), and root mean square error of test sets (RMSEP) were used to evaluate the performance of the regression model. The definitions were as follows:(1)$$R^{2}=1-\frac{\sum_{i=1}^{N}{\left({\hat{y}}_{i}-y_{i}\right)}^{2}}{\sum_{i=1}^{N}{\left({\hat{y}}_{i}-{\bar{y}}_{i}\right)}^{2}}$$
(2)$$\mathrm{RMSEC}\;\mathrm{or}\;\mathrm{RMSEP}\;=\sqrt{\frac{\sum_{i=1}^{N}{\left({\hat{y}}_{i}-y_{i}\right)}^{2}}{N}}$$
where ${\hat{y}}_{i}$ is the predicted adulteration level of the ith sample, $y_{i}$ is the true adulterated level of the ith sample, ${\bar{y}}_{i}$ is the average of $y_{i}$, and N is the number of samples.

### 4.6. Software

All the Raman spectral data pretreatments were performed using TheUnscrambler X14.1 software (CAMO, Oslo, Norway). All the calculations were performed using the R program (version 3.5.1). The Kknn package (version 1.3.1) was used for variable clustering, and the Caret package (version 6.0-82) was used for performing feature wavelength selections and machine learning modeling.

## 5. Conclusions

In this study, we evaluated the ability of a combined Raman spectroscopy and machine learning approach to rapidly detect the adulteration of Atlantic salmon using rainbow trout. A linear relation can be observed between the adulteration ratio of Atlantic salmon and the Raman spectra intensity. In this experiment, MSC was shown to be a better pretreatment method when compared with FD, SD, and SNV. GA was used to delete the irrelevant wavelengths, and KM was used to optimize the spectral bands. The Cubist method achieved the highest performance while modeling the spectra. Thus, the machine learning model developed in this study based on the MSC–GA–KM–Cubist method is an effective tool for quickly identifying the adulteration of Atlantic salmon meat.

## Figures and Tables

**Figure 1 molecules-24-02851-f001:**
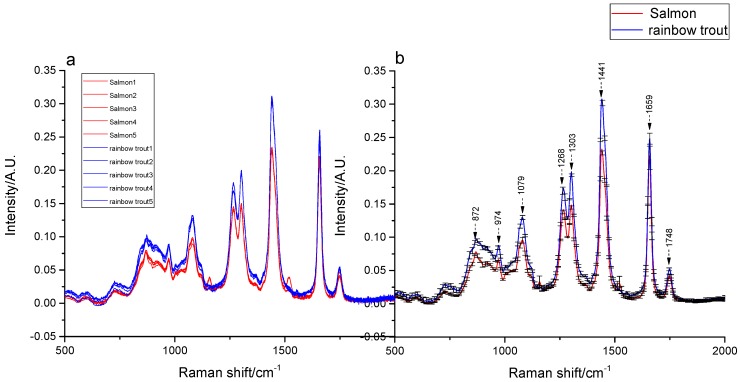
The Raman spectra of fat in Atlantic salmon and rainbow trout. (**a**) The five spectra of salmon and rainbow trout after baseline correction and multiple scattering correction (MSC). (**b**) The mean and standard deviation spectra of salmon and rainbow trout.

**Figure 2 molecules-24-02851-f002:**
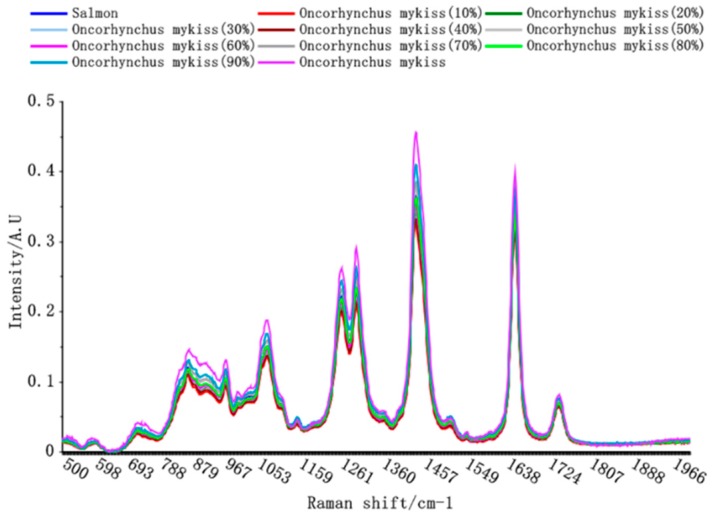
The Raman spectra observed in case of different proportions of rainbow trout adulteration in Atlantic salmon.

**Figure 3 molecules-24-02851-f003:**
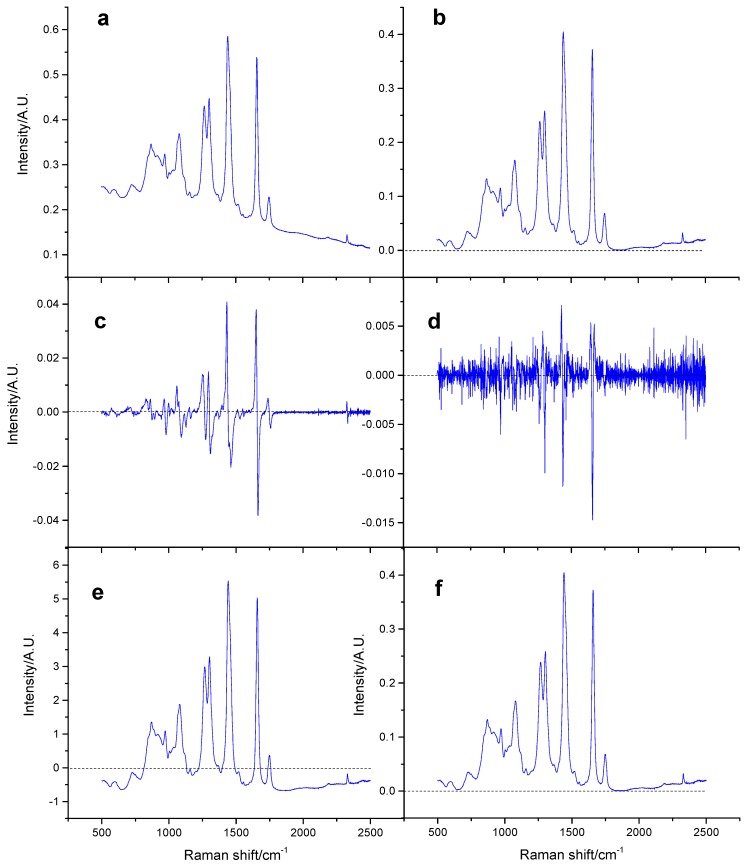
The Raman spectra of samples obtained using different pretreatments: (**a**) the original spectrum; (**b**) the spectrum after baseline fitting; (**c**) the spectrum after applying the first derivative; (**d**) the spectrum after applying the second derivative; (**e**) the spectrum after applying standard normal variate (SNV); (**f**) the spectrum after applying MSC.

**Figure 4 molecules-24-02851-f004:**
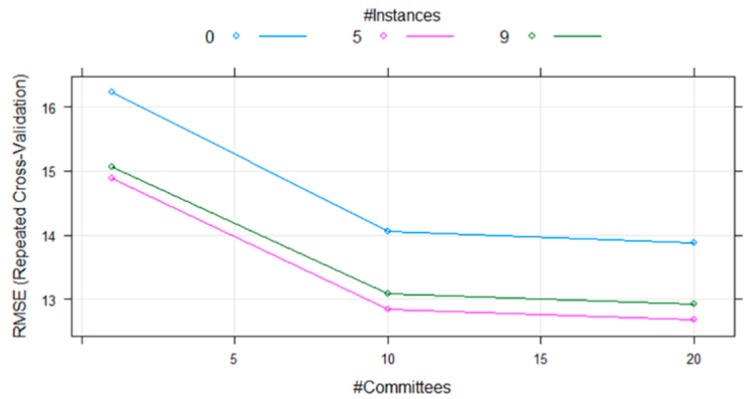
The cross-validated RMSE (Root Mean Square Error) curve for different commit sizes and instance numbers.

**Figure 5 molecules-24-02851-f005:**
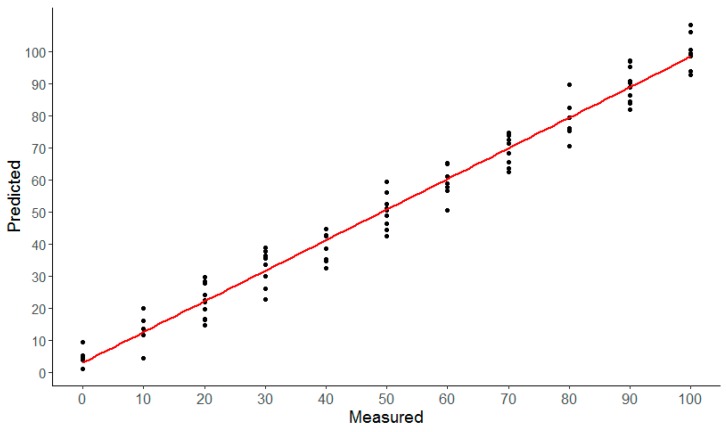
The predicted and true values of Atlantic salmon meat adulteration ratios based on the Cubist model in test sets.

**Table 1 molecules-24-02851-t001:** Raman spectral distribution of the Atlantic salmon fat.

Band/cm^−1^	Vibration Mode	Functional Groups	Intensity
1748	ν(C=O)	Ester (RC=OOR)	Weak
1659	ν(C=C)	Unsaturated band (cis RHC=CHR)	Strong
1441	δγ(C–H)	Methylene (CH_2_)	Strong
1303	δτ(C–H)	Methylene (CH_2_)	Medium
1268	δIP(=C–H)	Non-conjugated cis (RHC=CHR)	Medium
1079	ν(C–C)	–(CH_2_)_n_–	Medium
974	δ(=C–H)	Trans RHC=CHR	Medium
872	ν(C–C)	–(CH_2_)_n_–	Medium

**Table 2 molecules-24-02851-t002:** The partial least squares regression (PLSR) modeling results for four preprocessing methods.

Pretreatment Methods	Ncomp	Calibration Sets		Test Sets		
		RMSE (%)	R^2^	RMSEP (%)	R^2^P	MAE
NONE	10	14.79	0.79	17.27	0.70	13.19
FD	10	21.38	0.58	23.10	0.48	18.18
SD	10	29.26	0.19	30.15	0.12	25.11
SNV	10	13.66	0.82	13.28	0.81	10.49
MSC	9	13.68	0.82	13.32	0.81	10.57

MAE (Mean Square Error) denotes the average absolute error.

**Table 3 molecules-24-02851-t003:** Three feature wavelength selection methods based on the PLSR modeling results.

Dimension Reduction Methods	Number of Wavelengths	Calibration Sets		Test Sets		
		RMSE (%)	R^2^	RMSEP (%)	R^2^P	MAE
NONE	882	13.68	0.82	13.32	0.81	10.57
RFE–KM	75	14.47	0.79	14.93	0.77	12.24
GA–KM	431	14.36	0.80	13.34	0.81	10.69
SA–KM	322	14.55	0.79	13.84	0.80	11.11

MAE denotes the average absolute error.

**Table 4 molecules-24-02851-t004:** Performance comparison of different machine learning models.

Models	RMSE (%)		R^2^		MAE	
	Calibration Sets	Test Sets	Calibration Sets	Test Sets	Calibration Sets	Test Sets
PLS	14.36	13.34	0.80	0.81	11.18	10.69
Ridge	17.09	14.84	0.74	0.78	13.39	11.81
Enet	15.23	14.38	0.77	0.78	12.11	11.60
Rqlasso	15.72	14.92	0.76	0.77	12.46	11.94
Earth	16.30	16.84	0.74	0.71	12.93	13.14
Kknn	16.44	16.02	0.75	0.74	12.79	12.38
ParRF	15.91	14.87	0.77	0.79	12.92	11.91
Qrf	15.66	14.81	0.76	0.77	11.99	10.99
Rf	15.92	14.99	0.77	0.78	12.95	11.98
Ctree	21.74	22.71	0.55	0.48	17.09	16.95
Cubist	12.67	10.93	0.84	0.87	9.78	8.37
Glmboost	15.20	14.38	0.77	0.78	12.17	11.57
XgbTree	29.67	29.22	0.33	0.30	22.70	22.86
Msaene	15.33	14.39	0.77	0.78	12.37	11.69

MAE was the average absolute error.

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
