# Peer review of "Rapid Identification of Rainbow Trout Adulteration in Atlantic Salmon by Raman Spectroscopy Combined with Machine Learning"

_molecules, 2019, doi:10.3390/molecules24152851_

Round 1

Reviewer 1 Report

In this manuscript, the authors used Raman spectroscopy and machine learning to indentify the adulteration of rainbow trout in Atlantic salmon. They showed that the atlantic salmon meat adulteration was quantitatively analyzed on the basis of the Raman spectra in combination with the proper machine learning model. I think that this kind of study is unique and the manuscript may be publishable. However, there are several points to be modifed before it will be publishable, as listed below.

1. They employed several machine learning models, including Ridge, Enet, Earch and so on. But I could not understand what these models are like. The reference for each learning model should be added in Table 4 and section 4.4.2.

2.Section 4 (Material and Methods) should be placed after section 1 (Introduction)

3. Typos

Line 103: CH2 , 2 should be subscript.
Line 104 : CH Z stretching ?

Reviewer 2 Report

The manuscript presents Raman spectral method to identify and quantify rainbow trout adulteration in Atlantic salmon. Although the subject matter is interesting, non-existence of at least one unique fingerprint to distinguish one fish from the other is a major drawback of the methodology followed in this research. I would like the author to address my following concerns:

1. The Raman spectra of Atlantic salmon and rainbow trout fat in figure 1 shows that the peak position of both the fishes overlaps with each other. In Raman spectral analysis, the unique spectral fingerprint of a sample is needed for its identification. From figure 1, I am not convinced the Raman spectra can be used to identify/classify the two fishes.

2. Authors have claimed that the peak intensity difference is the basis for distinguishing between rainbow trout and Atlantic salmon. I would like to see 5 Raman spectra of same Atlantic salmon, 5 Raman spectra of same rainbow trout and 5 spectra each for different Atlantic salmon and rainbow trout. I also suggest the author plot mean and standard deviation spectra for the samples.

3. As the quantitative model is an equation which gives some number, please justify the result of your quantitative model. In Table 2 &3, RMSE and RMSEP values are high. In figure 5, taking an example of a sample at 0% concentration (i.e. pure sample), your prediction value is as high as 20%. Similarly, for 20% sample, the predicted concentrations range from 10% to 50%. This implies that the result obtained by the quantitative model is not justifiable.

4. I believe at least one non-overlapping Raman spectral peak should exist to identify components in the sample. Please refer published literature to justify otherwise.

Round 2

Reviewer 2 Report

The authors have made subsequent changes and have addressed my
queries. I would like to suggest authors to remove x-axis line
through the graph in all the figures, instead have x-axis at the
button. The manuscript can be accepted for publication after the
change.

Author Response

Response to Reviewer 2 Comments

 Point 1: The authors have made subsequent changes and have addressed my queries. I would like to suggest authors to remove x-axis line through the graph in all the figures, instead have x-axis at the button. The manuscript can be accepted for publication after the change.

Response 1: Thanks for pointing out that. These errors were corrected in the revised manuscript. But I don't know if we understand your meaning correctly.

With many thanks,

Truly yours,

Xingguo Tian